# A Rapid and Sensitive UHPLC–MS/MS Method for Determination of Chlorogenic Acid and Its Application to Distribution and Neuroprotection in Rat Brain

**DOI:** 10.3390/ph16020178

**Published:** 2023-01-24

**Authors:** Chongfei Bai, Xiaogang Zhou, Lu Yu, Anguo Wu, Le Yang, Jianping Chen, Xue Tang, Wenjun Zou, Jianming Wu, Linjie Zhu

**Affiliations:** 1Department of Chinese Materia Medica, School of Pharmacy, Chengdu University of Traditional Chinese Medicine, Chengdu 611137, China; 2School of Basic Medical Sciences, Southwest Medical University, Luzhou 646000, China; 3Department of Pharmacology, School of Pharmacy, Southwest Medical University, Luzhou 646000, China; 4Key Medical Laboratory of New Drug Discovery and Druggability Evaluation, Key Laboratory of Activity Screening and Druggability Evaluation for Chinese Materia Medica, Southwest Medical University, Luzhou 646000, China; 5Education Ministry Key Laboratory of Medical Electrophysiology, Southwest Medical University, Luzhou 646000, China; 6Chengdu Analytical Applications Center, Shimadzu (China) Co., Ltd., Chengdu 610023, China; 7School of Chinese Medicine, The University of Hong Kong, Hong Kong, China

**Keywords:** chlorogenic acid, pharmacokinetics, UHPLC–MS/MS, distribution, neuroprotection

## Abstract

Chlorogenic acid (5-CQA) is a phenolic natural product that has been reported to improve neurobehavioral disorders and brain injury. However, its pharmacokinetics and distribution in the rat brain remain unclear. In this study, we established a rapid and sensitive UHPLC–MS/MS method for the determination of 5-CQA in rat plasma, cerebrospinal fluid (CSF), and brain tissue to investigate whether it could pass through the blood–brain barrier (BBB) and its distribution in the rat brain, and a *Caenorhabditis elegans* (*C. elegans*) strain paralysis assay was used to investigate the neuroprotective effect of 5-CQA in different brain tissues. Chromatographic separation of 5-CQA and glycyrrhetinic acid (GA, used as internal standard) was completed in 0.5 min, and the full run time was maintained at 4.0 min. Methodological validation results presented a high accuracy (95.69–106.81%) and precision (RSD ≤ 8%), with a lower limit of quantification of 1.0 ng/mL. Pharmacokinetic results revealed that 5-CQA can pass through the BBB into the CSF, but the permeability of BBB to 5-CQA (ratio of mean AUC_0-∞_ of CSF to plasma) was only approximately 0.29%. In addition, 5-CQA can penetrate into the rat brain extensively and is distributed with different intensities in different nuclei. A *C. elegans* strain paralysis assay indicated that the neuroprotective effect of 5-CQA is positively correlated with its content in different brain tissues. In conclusion, our study for the first time explored the BBB pass rate and brain tissue distribution of 5-CQA administered via the tail vein by the UHPLC–MS/MS method and investigated the potential main target area of 5-CQA for neuroprotection, which could provide a certain basis for the treatment of nervous system-related diseases of 5-CQA.

## 1. Introduction

Chlorogenic acids are phenolic natural products composed of caffeic and quinic acids found in a variety of plants, fruits, vegetables, and coffee [1]. There are a number of isomers contained in it, of which 5-O-Caffeoylquinic acid (5-CQA, which was used in our study, Figure 1A) is the most abundant in plant and dietary sources [2,3]. 5-CQA is also one of the main active components in many traditional Chinese medicine injections, including Mailuoning injection, Shuanghuanglian injection, and Reduning injection [4,5,6]. Pharmacological studies have demonstrated that 5-CQA exhibits anti-inflammation, antioxidation, antiulcer, antiobesity, and antihypertension properties [7,8,9,10]. A growing number of studies have found that chlorogenic acid can effectively improve neurobehavioral disorders and brain injury in rats in recent years, showing obvious neuroprotective effects [11,12,13]. We also previously reported the brain-protective and antidepressant potential of 5-CQA through upregulating synapsin I expression to improve 5-hydroxytryptamine release [14]. Due to the importance of the pharmacokinetic characteristics of 5-CQA for further studies of its neuroprotective effects, an accurate method for quantifying 5-CQA in the brain is essential.

To date, there are few reports about the pharmacokinetics of 5-CQA. According to existing research, it is difficult to quantify 5-CQA at low concentrations due to the limitation of the instruments and equipment at the time, and the detection time is also long [15,16], or it may not be possible to quantify 5-CQA quickly and sensitively because of the need to detect multiple compounds at the same time [17,18,19]. Therefore, the 5-CQA quantitative method still needs to be optimized, such as by optimizing its quantitative limit and shortening its detection time. Moreover, there are no reports on the rate of chlorogenic acid blood–brain barrier passage, and the pharmacokinetics study of 5-CQA in the central nervous system (CNS) and its brain distribution remain unclear.

All drugs need to cross the blood–brain barrier (BBB) to exert the CNS efficacy, and studies have shown that drugs’ exposure in the CNS can be replaced by the detection of their concentrations in the cerebrospinal fluid (CSF) [20,21]. In the current study, the quantitative limit for 5-CQA was optimized, as was its detection time. In addition, a rapid and sensitive ultra-high-performance liquid chromatography–tandem mass spectrometry (UHPLC–MS/MS) method was established to further perform a study on the pharmacokinetics and brain distribution of 5-CQA in rat plasma and CSF and obtain its pass ratio through the BBB into CSF. Moreover, the concentration of 5-CQA in different brain tissues was measured, and a *Caenorhabditis elegans* (*C. elegans*) model, a common model used to investigate drug toxicity and neuroscience, was used to compare its neuroprotective effect [22]. Our study provided a superior method for the determination of 5-CQA in the CSF and brain tissue of rats and identified its potential main target area of brain-protective effects. The results may enhance the understanding of 5-CQA in neuroprotection and provide potential guidance for the clinical use of related traditional Chinese formulas.

## 2. Results

### 2.1. LC–MS/MS Analysis

Both positive and negative modes were used to make ESI conditions as optimal as possible for determining 5-CQA and GA. 5-CQA and IS displayed more and stronger signals in negative mode than in positive mode. In the end, a negative ion was applied. For the purpose of establishing a quantitative method for MS, full scans of precursor ions and the production of 5-CQA and IS were conducted to affirm their precursor ions and the corresponding products. Then, the MS parameters, examples including cell accelerator voltage, collision energy, and fragment, were optimized to produce the greatest relative abundance of precursor and products. The quantitation was performed by using the ion pair including *m*/*z* 353.15→191.15 for 5-CQA and *m*/*z* 469.20→355.15 for IS in MRM mode (Figure 2). The key point was to choose an appropriate IS, which could enhance the method performance. Since the final GA had similar chromatographic behavior, ionization response, and extraction efficiency as 5-CQA, it was selected as the IS.

### 2.2. Method Validation

Rat CSF, plasma, and brain homogenate samples were used to evaluate the selectivity of the analytical method for endogenous plasma matrix. The retention times of 5-CQA and GA were 1.7 min and 3.43 min, respectively. Representative MRM chromatograms were obtained from blank rat CSF, plasma, and brain homogenate samples, spiked rat CSF and plasma samples with 5-CQA and GA, and rat CSF and plasma samples after a 5 min tail vein injection of 5-CQA. As shown in Figure 3, the peaks of 5-CQA and IS were detected with good peak shape and excellent resolution, and the 5-CQA retention time was not affected by endogenous substances. By plotting the peak area ratio of 5-CQA to GA against the plasma, CSF, and brain homogenate concentrations, calibration curves of 5-CQA in rat plasma, CSF, and brain homogenates were generated. Rat plasma, CSF, and brain homogenates were analyzed with linear regression of the peak area ratios versus concentration over the concentration range of 2–400 ng/mL. The linear 5-CQA in rat plasma, CSF, and brain homogenate were y = 0.00413x + 0.0073 (R^2^ = 0.998, plasma), y = 0.00406x + 0.0078 (R^2^ = 0.999, CSF), and y = 0.01510x + 2.1116 (R^2^ = 0.991, brain), respectively. The LLOQ of 5-CQA was 1.0 ng/mL in all biological samples. In the current study, QC samples at different 5-CQA concentrations were analyzed on three different days and on the same day to determine interday and intraday precision and accuracy. The results are listed in Table 1, indicating that the interday and intraday precision of plasma, CSF, and brain ranged from 2.22% to 8.00%, 2.40% to 7.70%, and 4.19 to 6.89%, respectively. The accuracy of plasma samples ranged from 95.69% to 106.81% and 99.73% to 106.73% for the CSF sample, and 95.78 to 101.43% for brain homogenates. All the above results were within a range of 100 ± 15%, which is within the FDA standard limits [23]. The matrix effects at low, middle, and high concentrations of 5-CQA and GA are shown in Table 2. The matric effects were in the range of 85–115%. The results identified that the current determination method was not affected by matrix effect. The stability of 5-CQA at different concentrations (2, 50, and 400 ng/mL) in rat biological samples is shown in Table 3. The RSD of the stability of all biological samples ranged from 1.99% to 9.32% in all stability assays. The results suggested that 5-CQA showed no significant degradation during the process of analysis.

### 2.3. Pharmacokinetics Study

The average concentration–time profiles of 5-CQA in the plasma and CSF are shown in Figure 4. The main pharmacokinetic parameters of 5-CQA are shown in Table 4. The results showed that 5-CQA can be detected simultaneously in plasma and CSF within 5 min after intravenous injection, which suggests that 5-CQA can cross the BBB. According to DAS 3.0, we observed that only a very small percentage of 5-CQA can penetrate into the CSF. The maximum plasma concentration (C_max_), area under the concentration–time curve (AUC_0-t_), and AUC_0-∞_ in plasma were nearly 287 times (42,545.60 ng/mL vs. 148.35 ng/mL), 246 times (34,985.91 ng/h/mL vs. 142.30 ng/h/mL), and 343 times (60,509.72 ng/h/mL vs. 176.81 ng/h/mL), respectively, and the BBB permeability of 5-CQA was approximately 0.29% (ratio of mean AUC_0-∞_ of CSF to plasma, based on the approach used in previous studies) [24]. Another noteworthy point was that the t_1/2_ of 5-CQA was faster in CSF than in plasma (0.44 h vs. 1.24 h).

### 2.4. Brain Nuclei Distribution and Neuroprotective Effect

The concentrations of 5-CQA in the rat Hip, Cor, Hyp, and BS after vein injection of 1 dose of 50 mg/kg of 5-CQA are shown in Figure 5. The results showed that 5-CQA could be detected at different concentrations in Hip, Hyp, Cor, and BS at 5 min after intravenous injection, and the concentration of 5-CQA in the BS was the highest. The concentration of 5-CQA in BS demonstrated a significant trend toward decreasing over time. Overall, the variation trend of 5-CQA concentration in Hyp was completely opposite, showing an increasing trend with time. In addition, the concentration of 5-CQA in both Hip and Cor fluctuates with time. Notably, 5-CQA reached maximum concentrations in Hip and Cor at 30 and 120 min, respectively.

The effect of different brain nuclei homogenate on the paralysis of CL4176 worms was determined to further investigate the main target areas of the 5-CQA neuroprotective effect. As shown in Figure 6 (visualization of worm paralysis in different groups is shown in the Appendix A), all brain nuclei homogenates significantly reduced temperature-induced paralysis of worms, and the BS group provided the best protection.

## 3. Discussion

Chromatographic separation conditions play a pivotal role in peak shape, sensitivity, and run time, and a good and stable detection method is the premise of the study of pharmacokinetics [25]. Compared to methanol, acetonitrile was selected to precipitate protein during the whole experiment since it precipitated more completely and had less interference. In addition, several mobile phase conditions, such as acetonitrile–water, methanol–water, and other solutions, including acetic acid, formic acid, and ammonium acetate, were tested. In the end, 0.05% formic acid in water and 5 mM ammonium formate in acetonitrile were selected as the optimal solvent composition to obtain a better peak shape and flatter baseline.

Compared with the reported detection of chlorogenic acid, we improved its analytical method by UHPLC–MS/MS [26]. In detail, we shortened the analysis time of chlorogenic acid to 4 min and lowered the LLOQ to 1 ng/mL, which may save time for future analysis of chlorogenic acid. In addition, we investigated the distribution of chlorogenic acid in brain tissue using UHPLC–MS/MS and initially identified the potential target areas of 5-CQA performing neuroprotective effects for the first time. Unfortunately, our study mainly concentrated on the distribution of 5-CQA in brain tissue and its main target area for neuroprotection, as our previous research found that 5-CQA can effectively delay the progression of Parkinson’s disease [27]. The main metabolites of 5-CQA were not detected simultaneously in this study, and thus, we could not prove whether it is 5-CQA or its metabolite that mainly exerts neuroprotective effects. However, it is definitely the case that investigating the material basis of 5-CQA in neuroprotective effects is our further focus, and we will certainly establish a UHPLC–MS/MS approach for simultaneous detection of 5-CQA and its metabolites.

To our knowledge, more than 30 chlorogenic acid metabolites have been detected, among which caffeic acid, dihydrocaffeic acid, m-coumaric acid, and quinic acid have been reported to have pharmacological activity [28,29]. It has been reported that caffeic acid has significant antioxidant activity, dihydrocaffeic acid confers significant protection against oxidative damage, and m-coumaric acid was found to promote neurite outgrowth in hippocampal neuronal cells [30,31,32]. Although there are many studies on 5-CQA and its metabolites, it is still unclear in which brain tissues they are mainly distributed to provide neuroprotective effects, which is why we plan to carry out further studies as described above.

Chlorogenic acid is a widely distributed natural substance in a variety of plants, vegetables, and fruits [33]. Although the route of administration in reported studies on chlorogenic acid was also mostly oral, intravenous injection was used in the present study [34]. 5-CQA has been reported to have poor penetration across lipophilic membrane barriers, and its oral bioavailability is even less than 1% [35]. To accomplish the objectives of our study, it was necessary to ensure that the concentration of 5-CQA entering the brain tissue was adequate and could be accurately quantified. Due to the above considerations, we finally selected tail vein injection of a high dose of 5-CQA after pretesting. Moreover, the pharmacokinetic parameters of 5-CQA, including C_max_, T_max_, AUC, and t_1/2_, were different with different administration methods, dosages, and test animals. Studies showed that the absolute bioavailability of 5-CQA was only 4.8% after oral 50 mg/kg of 5-CQA, which was calculated to be concordant with our results [36]. Therefore, how to accurately quantify chlorogenic acid in different brain regions while conforming to the daily intake and clinical dose of chlorogenic acid is what we need to address in our next study.

## 4. Materials and Methods

### 4.1. Chemicals and Reagents

5-CQA (C_16_H_18_O_9_, MW = 354.31, purity, 98%) was obtained from Nanjing King Bamboo Biological Technology Co., Ltd. (Nanjing, China). Glycyrrhetinic (GA, C_30_H_46_O_4_, MW = 470.69, purity, 99%) was obtained from the National Institute for the Control of Pharmaceutical and Biological Products (Beijing, China) as the internal standard (IS) for UHPLC–MS/MS and pharmacokinetic analysis of 5-CQA. The chemical structures of 5-CQA and GA are shown in Figure 1. LC–MS grade acetonitrile was purchased from Merck Company (Darmstadt, Germany). HPLC grade formic acid was purchased from Ann Spectrum Company (Shanghai, China), and HPLC grade ammonium formate (purity ≥ 98%) was purchased from Shanghai Hongrui Chemical Co., Ltd. (Shanghai, China). Ultrapure water used throughout the experiment was made by a Millipore water purification system (Bedford, MA, USA). All of the other reagents and solvents were of analytical grade.

### 4.2. Instrumentation and Analytical Conditions

The analysis was performed on a Shimadzu LCMS-8040 UHPLC system (Shimadzu, Kyoto, Japan) that included two LC-30AD infusion pumps, a CTO-30A column oven, a SIL-30AC automatic sampling device, a DGU-20A5 Solvent Degasser, CBM-20A system controller, an LCMS-8040 triple quadrupole mass spectrometer, and Lab Solutions station (Ver. 5.75, Shimadzu Corporation, Kyoto, Japan) chromatography workstation in conjunction with a Shim-pack XR-ODS III Column (2.0 mm I.D. × 50 mm L., 1.6 μm). The gradient elution settings of the mobile phase mixture (A) water-formic acid (100:0.05, *v*/*v*) and (B) 5 mM ammonium formate-acetonitrile with a gradient elution were set as follows: 5–5% B at 0.0–0.8 min, 5–95% B at 0.8–1.8 min, 95–95% B at 1.8–2.9 min, 95–5% B at 2.9–3.0 min, and 5–5% B at 3.0–4.0 min. The flow rate was maintained at 0.35 mL/min, and the autosampler was at 25 °C during the whole process. The injection volume was 10 µL, and the total chromatographic time was 4.0 min.

The mass spectrometer was used in negative ion ionization mode. The voltage ion source interface was −3.0 kV, and the desolvation line temperature was 250 °C. The heating module temperature was 450 °C. Multiple reaction monitoring (MRM) mode was used for quantification by observing the precursor ion to produce ion transitions of *m*/*z* 353.15→191.15 for 5-CQA and *m*/*z* 573.15→531.15 for GA (Figure 2). High-purity nitrogen (N2) was employed as the atomizing gas, and N2 was served as the drying gas with a flow rate of 2.5 L/min and 12.0 L/min, respectively.

### 4.3. Animals

Male Sprague–Dawley rats (8–10 weeks of age, body weight: 250 ± 50 g, license number: SCXK2013–17) were obtained from Southwest Medical University (Luzhou, Sichuan, China). SD rats had free access to diet and water and adapted to 5 rats in each cage under RH 55 ± 5% and 22 to 25 °C with a 12 h light/dark cycle. Rats were adaptively fed for 7 days and then fasted for 12 h but always had free access to water before experiments. All operations on rats complied with guidelines endorsed by the Laboratory Animal Ethics Committee of Southwest Medical University (Luzhou, China).

The AD worm strain CL4176, dvIs27 [myo-3p::A-Beta (1–42)::let-851 3’UTR) + rol-6(su1006)] X used in this study was obtained from the Caenorhabditis Genetics Center (CGC), maintained on NGM plates, and fed with Escherichia coli OP50 at 16 °C unless otherwise noted.

### 4.4. Samples Preparation

5-CQA (1.07 mg) was dissolved in pure water at a concentration of 1.00 mg/mL as a stock solution, which was further attenuated with pure water to obtain a series of concentrations (10–500 ng/mL) as a standard working solution.

GA (1.05 mg) was dissolved in 1.05 mL methanol to obtain a stock solution with a concentration of 1.00 mg/mL, which was then attenuated with methanol to obtain a working solution with a concentration of 1000 ng/mL. We freshly prepared all of the solutions before experiments.

Biological calibration standards for 5-CQA were produced by spiking a suitable amount of the working solution of 5-CQA and IS into 45 µL of blank plasma or 89 µL of CSF sample and brain homogenate. The solution was thoroughly mixed by vortexing for 30 s, and then 200 µL 5% formic acid-acetonitrile was pipetted and vortexed for 1 min. After the supernatant was used for detection by centrifugation at 21,913× *g* for 10 min, the quality control (QC) samples were made at concentrations of 5, 50, and 500 ng/mL in the same way as the calibration standards were prepared. All solutions were stored at −20 °C.

Frozen samples of plasma and CSF were placed away from light at room temperature for 30 min before analysis. After the samples were thawed completely, the appropriate amount of IS working standard solution was added into 45 µL plasma or CSF sample and thoroughly vortexed for 30 s. To precipitate protein, 200 µL of 5% formic acid-acetonitrile was added and vortexed for 1 min in all samples. After centrifugation at 21,913× *g* for 10 min, the supernatants were collected for detection.

Rats were sacrificed by decapitation after anesthesia, and the brain tissues were completely peeled off. Then, the hippocampus (Hip), cortex (Cor), hypothalamus (Hyp), and brainstem (BS) were divided, washed using normal saline, gently blotted on filter paper, and weighed. Hip, Cor, Hyp, and BS samples were homogenized in 2 volumes of normal saline by using a homogenizer. The appropriate amount of IS working standard solution was added to 45 µL brain homogenate and thoroughly vortexed for 30 s. Then, 200 µL of 5% formic acid-acetonitrile was added and it was vortexed for 1 min to precipitate protein. After centrifugation at 21,913× *g* for 10 min, the supernatants were gathered for detection.

### 4.5. Method Validation

The specificity was confirmed by comparatively analyzing the difference between blank plasma, CSF samples, or brain homogenates, and blank plasma, CSF samples, or brain homogenates spiked with 5-CQA and GA.

The calibration curve of 5-CQA in plasma was constructed by matching the ratio of peak area of 5-CQA to IS to a linear equation with the corresponding concentrations. Calibration standards containing eight different concentrations of 5-CQA (1.0, 2.0, 5.0, 20, 50, 100, 250, and 500 ng/mL) were measured to obtain the calibration curves. The linearity of each calibration curve was obtained by plotting the linear regression of the peak area ratios of 5-CQA/IS against the 5-CQA concentrations spiked in biological samples. The lowest concentration of 5-CQA on the calibration curve was defined as the lower limit of quantitation (LLOQ). Meanwhile, the precision of this method was represented as a percentage of relative standard deviation (RSD), and the accuracy was expressed as a relative error (RE), accessed by comparing samples in five replicates.

The precision and accuracy of the current analytical method were validated at three concentration levels of 5-CQA (2, 50, and 400 ng/mL). The intraday precision and accuracy were evaluated by detecting QC samples in a single day, and the interday precision and accuracy were evaluated by analyzing the QC samples for more than 3 days in a row.

Three different concentrations levels (2, 50, and 400 ng/mL) of 5-CQA were used to assay the matrix effect by analyzing the peak areas achieved from the standards with those obtained from blank biological samples spiked with 5-CQA and GA. Briefly, 45 μL blank rat CSF, plasma samples, or brain homogenates were spiked with 10 μL of IS standard working solution and three different 5-CQA standard working solutions (final concentrations: 2, 50, and 400 ng/mL), which were further precipitated with 200 μL ammonium formate-acetonitrile (5 mM) and vortexed thoroughly. After centrifugation at 21,913× *g* for 10 min, supernatant was collected for analysis. The QC samples were prepared as described above. All the prepared samples were analyzed using UHPLC–MS/MS.

The samples used for the investigation of stability were prepared according to the calibration standards. A total of three concentration levels (2, 50, and 400 ng/mL) of 5-CQA were used for biological samples, and each concentration had five replicate samples. The samples were placed away from light during the experiment. The postpreparative storage stability was evaluated after the prepared QC samples were stored in a 4 °C refrigerator for 18 h. The short-term stability was determined after the QC samples were stored at indoor temperature for 2 h. The freeze–thaw stability was determined by freezing the samples at −80 °C for at least 12 h, then defrosting them at indoor temperature three times repeatedly. The stability of 5-CQA was assayed by determining the concentrations of 5-CQA in the samples using UHPLC–MS/MS following the calibration curve of 5-CQA in plasma or CSF samples.

### 4.6. Pharmacokinetics Study

Pentobarbital sodium (30 mg/kg) was intraperitoneally injected into five rats to anesthetize them. A gum-elastic catheter consisting of polyethylene and silicone rubber tubing was catheterized into the right extracervical vein. These rats were then fed one rat per cage overnight to completely wake up. On the second day, the rats were then secured to a stereotaxic apparatus for cannulating their cisterna magna with a 25-gauge needle attached to a 1.5 cm Eppendorf tube as described previously [25]. After the cannula was inserted into the cisterna magna, the rats were given 50 mg/kg 5-CQA dissolved in physiological saline by tail vein injection as previously described [26]. Blood and CSF samples were gathered by catheters at various time points (pretreatment, 5, 10, 20, 30, 45 min, 1, 1.5, 2, 3, 4, 6, and 8 h after 5-CQA administration); 300 μL blood and 20 μL CSF were collected each time. Sterile filtered 0.9% saline was given to the rats by oral administration to supplement the blood loss with the amount of water replacement equal to the amount of blood drawn. Finally, plasma samples were gathered in anticoagulant Eppendorf tubes and centrifuged at 1006× *g* for 5 min, and the supernatants were collected for further analysis.

### 4.7. Brain Nuclei Distribution

Twenty rats were divided randomly into four groups, with five rats per group. Rats were administered 50 mg/kg 5-CQA dissolved in normal saline by tail vein injection. After injection, the rats were slain at 5, 30, 120, and 240 min, and the Hip, Cor, Hyp, and BS were excised. The brain nuclei samples were weighed and stored at −80 °C until treatment.

CL4176 is a transgenic nematode that can express human Aβ1-42 protein in its muscle tissue induced by temperature and then rapidly become paralyzed, and it plays an important role in studying the pathogenesis of Alzheimer’s disease and the drugs for its treatment [37,38]. A paralysis assay in CL4176 worms was performed as previously described. In short, the synchronized L1 larvae of CL4176 worms (100–150 worms per condition) were transferred to NGM plates which contained the homogenate of brain tissue of rats administered with or without 5-CQA. The NMG plates were incubated at 16 °C for 36 h and then changed to 23 °C to induce the expression of human Aβ1-42 protein in the muscle of CL4176 worms, which ultimately led to the paralysis of nematodes. Then, the worms were scored for paralysis after the temperature shifted for 36 h, and nematodes were observed to have “halos” around their heads, which removed bacteria or moved their head only or even did not move when they were touched by a platinum worm to pick the nematodes and were regarded as paralyzed. The assay results were obtained from more than three independent experiments.

### 4.8. Statistical Analysis

Data are expressed as the mean ± standard deviation (mean ± SD). Pharmacokinetic parameters were calculated by DAS 3.0 software. Statistical analysis was performed with GraphPad Prism (Prism 5.0, San Diego, CA, USA) using a two-tailed Student’s *t* test, and *p*-values less than 0.05 were considered as statistically significant. Representative photographs and videos of paralyzed CL4176 worms were taken under a Leica M205FA stereomicroscope (Leica, Wetzlar, Germany). Bandicam software (Bandicam Company, Seoul, Korea) was used for analysis. The graphical scheme of study approach is shown in Figure 7.

## 5. Conclusions

A simple, sensitive, and rapid UHPLC–MS/MS method was developed in our study and applied to investigate the pharmacokinetic characteristics of 5-CQA, by which we calculated the BBB permeability of 5-CQA administered via tail vein and investigated its distribution in the brain and the potential major targets of neuroprotection for the first time. The results indicated that 5-CQA could rapidly cross the BBB, but its penetration rate was only approximately 0.29%. Noticeably, the t_1/2_ of 5-CQA in rat CSF was faster than that in plasma. Moreover, 5-CQA can penetrate extensively into rat brain and is distributed in various nuclei with different intensities. This study provided worthy data for research on 5-CQA in neurodegenerative diseases and might be helpful in revealing the mechanism of 5-CQA in exerting neuroprotective effects. Unfortunately, we were not able to confirm whether it is 5-CQA or its metabolite that exerted neuroprotective effects due to the limitations of the experimental conditions.

## Figures and Tables

**Figure 1 pharmaceuticals-16-00178-f001:**
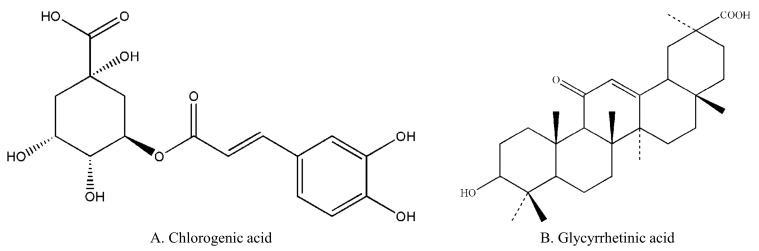
Chemical structures of chlorogenic acid (**A**) and glycyrrhetinic acid (**B**).

**Figure 2 pharmaceuticals-16-00178-f002:**
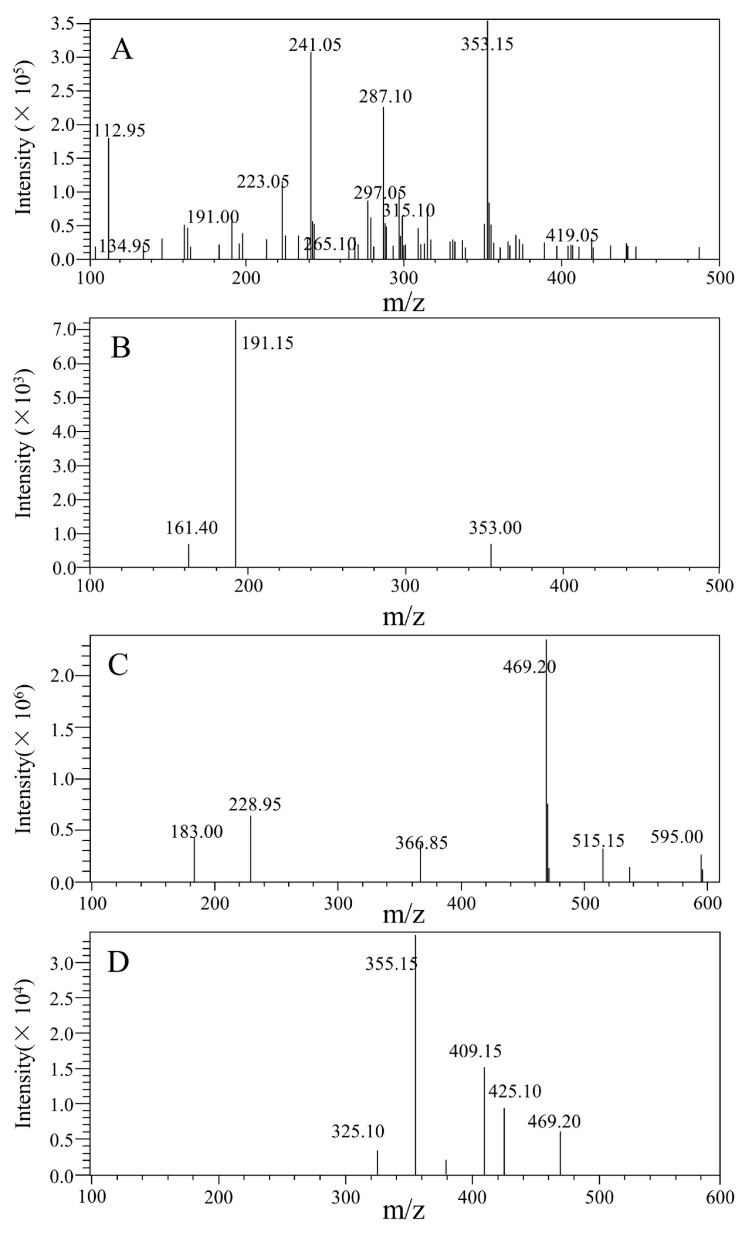
Full-scan precursor ion and production spectra of 5-CQA and GA. (**A**) Full-scan precursor ion spectra of 5-CQA; (**B**) full-scan product ion spectra of 5-CQA; (**C**) full-scan precursor ion spectra of GA; (**D**) full-scan product ion spectra of GA.

**Figure 3 pharmaceuticals-16-00178-f003:**
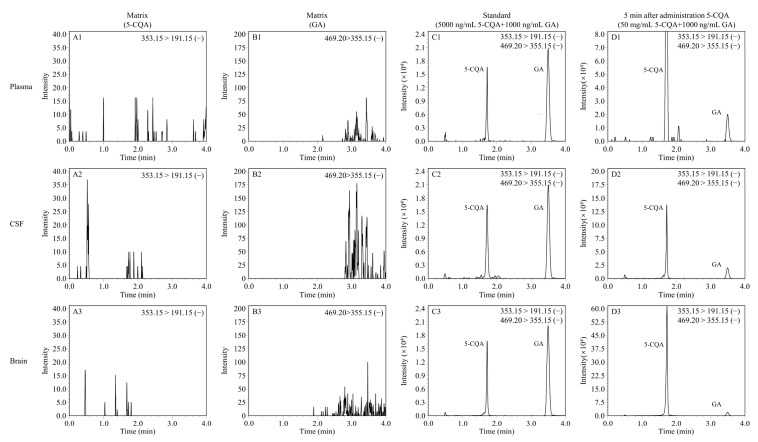
Representative MRM chromatograms of 5-CQA and GA in rat plasma, CSF, and brain samples. (**A1**) Blank rat plasma sample without 5-CQA; (**B1**) blank rat plasma sample without GA; (**C1**) blank rat plasma sample with 500 ng/mL 5-CQA and 1000 ng/mL GA; (**D1**) rat plasma sample collected at 5 min after tail vein injection of 50 mg/kg 5-CQA; (**A2**) blank rat CSF sample without 5-CQA; (**B2**) blank rat CSF sample without GA; (**C2**) blank rat CSF sample with 500 ng/mL 5-CQA and 1000 ng/mL GA; (**D2**) rat CSF sample collected at 5 min after tail vein injection of 50 mg/kg 5-CQA; (**A3**) blank rat brain homogenate sample without 5-CQA; (**B3**) blank rat brain homogenate sample without GA; (**C3**) blank rat brain homogenate sample with 500 ng/mL 5-CQA and 1000 ng/mL GA; (**D3**) rat brain homogenate sample collected at 5 min after tail vein injection of 50 mg/kg 5-CQA.

**Figure 4 pharmaceuticals-16-00178-f004:**
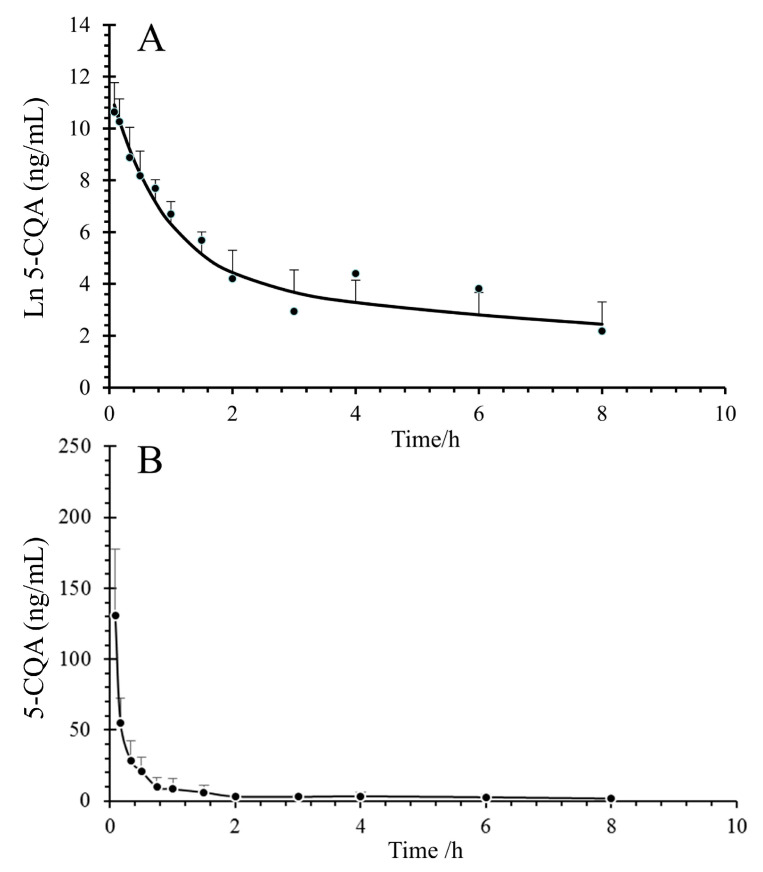
Concentration–time curves of 5-CQA after tail vein injection of 50 mg/kg 5-CQA. (**A**) Concentration–time curve of rat plasma; (**B**) concentration–time curve of rat CSF.

**Figure 5 pharmaceuticals-16-00178-f005:**
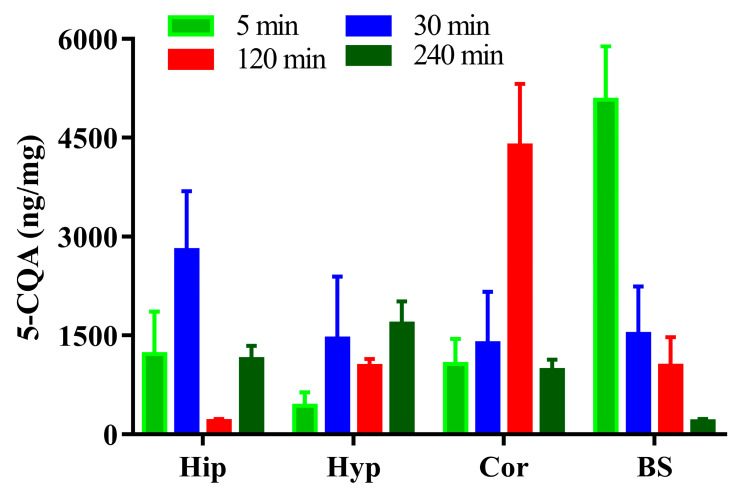
Concentration of 5-CQA in the rat hippocampus, hypothalamus, cortex, and brainstem after vein injection of 50 mg/kg 5-CQA.

**Figure 6 pharmaceuticals-16-00178-f006:**
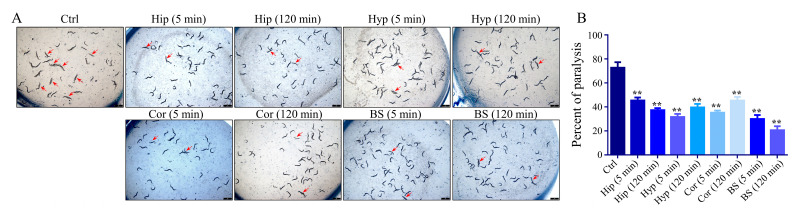
Protective effect of hippocampus, cortex, hypothalamus, and brainstem homogenates on paralyzed CL4176 worms. (**A**) Representative images were captured under a Leica M205FA stereomicroscope at 36 h; (**B**) quantification of paralyzed CL4176 worms at 36 h treatment with four brain tissue homogenates. (** *p* < 0.01 vs. control group; *n* = 3.)

**Figure 7 pharmaceuticals-16-00178-f007:**
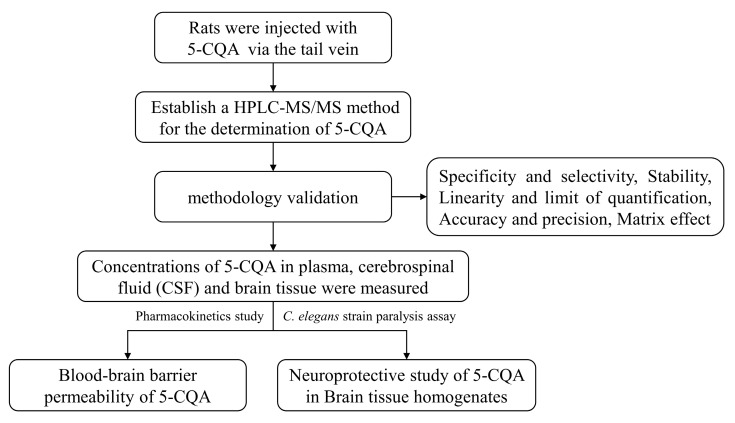
The graphical scheme of our study approach.

**Table 1 pharmaceuticals-16-00178-t001:** Inter- and intraday precision and accuracy for 5-CQA in rat biological samples (*n* = 5).

Biological Matrices	Spiked (ng/mL)	Interday (RSD, %)	Intraday (RSD, %)	Accuracy (mean ± SD, %)
Plasma	2	8.00	5.90	95.69 ± 6.84
	50	4.70	3.81	101.55 ± 4.77
	400	3.81	2.22	106.81 ± 4.07
CSF	2	7.70	7.70	99.73 ± 7.68
	50	3.56	3.56	106.73 ± 3.80
	400	2.40	2.40	104.92 ± 2.51
Brain	2	5.31	6.89	96.25 ± 8.09
	50	5.67	4.19	101.43 ± 5.87
	400	6.40	5.83	95.78 ± 4.53

**Table 2 pharmaceuticals-16-00178-t002:** Matrix effect of 5-CQA in the plasma, CSF, and brain homogenates of rats (*n* = 5).

Analyte	Spiked (ng/mL)	Matrix Effect (Mean ± SD, %)	Matrix Effect (Mean ± SD, %)	Matrix Effect (Mean ± SD, %)
5-CQA	2	97.32 ± 7.97	107.08 ± 8.76	98.43 ± 6.98
	50	95.82 ± 5.98	105.06 ± 6.56	101.45 ± 3.89
	400	93.77 ± 4.09	103.14 ± 4.50	105.41 ± 5.62
IS	1000	102.80 ± 3.13	101.93 ± 2.78	103.98 ± 3.14

**Table 3 pharmaceuticals-16-00178-t003:** Stability of 5-CQA in the plasma, CSF, and brain homogenate samples of rats (*n* = 5).

Matrix	Nominal (ng/mL)	Postpreparative (4–8 °C, 18 h)		Short-Term (25 °C, 2 h)		After Three Freeze-Thaw Cycles	
		Measured (ng/mL)	RSD (%)	Measured (ng/mL)	RSD (%)	Measured (ng/mL)	RSD (%)
Plasma	2	2.21 ± 0.17	7.52	2.21 ± 0.17	7.52	2.21 ± 0.17	7.52
	50	52.81 ± 3.06	5.79	52.81 ± 3.06	5.79	52.81 ± 3.06	5.79
	400	406.82 ± 13.06	3.21	406.82 ± 13.06	3.21	406.82 ± 13.06	3.21
CSF	2	2.16 ± 0.11	5.09	1.91 ± 0.13	6.81	1.84 ± 0.11	5.98
	50	51.77 ± 1.03	1.99	50.09 ± 3.01	6.01	47.35 ± 4.35	9.19
	400	404.00 ± 18.50	4.58	387.67 ± 36.13	9.32	364.29 ± 30.46	8.36
Brain	2	1.98 ± 0.14	3.88	2.08 ± 0.31	6.78	2.13 ± 0.09	2.51
	50	50.83 ± 1.14	1.99	51.87 ± 2.04	2.45	51.23 ± 3.24	2.45
	400	405.87 ± 16.37	4.31	407.61 ± 26.01	3.78	405.31 ± 16.43	5.21

**Table 4 pharmaceuticals-16-00178-t004:** The main pharmacokinetic profile of 5-CQA in the plasma and CSF of rats after vein injection of a single dose of 50 mg/kg (*n* = 5).

Parameter	In CSF	In Plasma
C_max_ (ng/mL)	148.35 ± 43.13	42,545.60 ± 3040.20
t_1/2_ (h)	0.44 ± 0.27	1.24 ± 0.96
AUC_0-t_ (ng/h/mL)	142.30 ± 67.33	34,985.91 ± 17,066.36
AUC_0-∞_ (ng/h/mL)	176.81 ± 77.44	60,509.74 ± 39,713.01

## Data Availability

Data are contained within the article and Appendix A.

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
