# Peer review of "A Rapid and Sensitive UHPLC–MS/MS Method for Determination of Chlorogenic Acid and Its Application to Distribution and Neuroprotection in Rat Brain"

_pharmaceuticals, 2023, doi:10.3390/ph16020178_

Round 1

Reviewer 1 Report

In this manuscript, the author descripted a UHPLC–MS/MS method for the determination of chlorogenic acid in rat plasma, cerebrospinal fluid (CSF) and brain tissue and its application to distribution and neuro-protection in rat brain. The method was validated in linearity, precision, accuracy, selectivity, lower limit of quantitation, matrix effects, stability, et al. Overall, the paper is well-written to the point with thorough literature review.

Author Response

Thank you for your recognition of the performance of this work and the quality of the article. We have also modified and improved it according to the comments of other reviewers. We tried our best to improve the manuscript and made some changes to the manuscript. These changes will not influence the content and framework of the paper. And here we marked in red in the mainly revised paper. We appreciate for your warm work earnestly and hope that the correction will meet with approval. Once again, thank all of you very much for your painstaking review.

Reviewer 2 Report

This work developed a rapid and sensitive UHPLC–MS/MS method for determining chlorogenic acid in rat plasma, cerebrospinal fluid, and brain tissue, to investigate the pharmacokinetics. However, there are some points to be revised:

-          One author is affiliated with a commercial company, Is conflict of interest must be mentioned in the Conflict of interest section?

-          Caenorhabditis elegans (C. elegans) must be italicized.

-          Please check C16H18O9 and C30H46O4; numbers must be subscripted.

-          Please check R2; numbers must be superscripted.

-          For the animal study, how long of acclimatization of the animal prior test?

-          Please check the parameter term and unit of penetration rate was only 0.29%

-          Is AUC0-t mean AUC0-8?

-          Why the trend line in Figures 4A and 4B are different? Is the error bar with the dot symbol rather than the line for Figure 4A?

-          Lines 154-165; Please check the result in the text mentioned for Fig. 5. Are there any mistakes?

-          Which is the FDA guideline the author followed? Please add a reference.

-          Introduction; the authors mentioned that ref. 12-14 had long analysis time and complex sample processing procedures. In my opinion, I am not satisfied with this issue. Ref 12-14 are analysis 5-CQA in plant extracts that are very complex than the analysis of one or two pure compounds. I strongly recommend improving this paragraph.

- Similarity from Turnitin found that 34%, it will be nice if the authors rewrite some sentences to decrease the similarity index.

Author Response

Thank you for your comments concerning our manuscript (ID: pharmaceuticals-2111736). Those comments are all valuable and very helpful for revising and improving our paper. We have studied comments carefully and tried our best to improve the manuscript and made some changes to the manuscript. These changes will not influence the content and framework of the paper. And here we marked in red in the mainly revised paper. We appreciate for your warm work earnestly and hope that the correction will meet with approval. Once again, thank all of you very much for your painstaking review. The main corrections in the paper and the response to your comments are attached.

Round 2

Reviewer 2 Report

All comments are answered and addressed. It can be accepted in this present form.

Author Response

Thank you for your approval and positive comments on the revised manuscript. In addition, we sincerely appreciate your patience and careful review of the manuscript to improve its quality. Without your professional comments, our manuscript would not have met the requirements for publication. The questions you pointed out and the helpful comments will be very valuable for the design and writing of our future articles.

Thanks again for your time and effort on this manuscript.